# Animal Forensic Genetics

**DOI:** 10.3390/genes12040515

**Published:** 2021-04-01

**Authors:** Adrian Linacre

**Affiliations:** College of Science & Engineering, Flinders University, Adelaide, SA 5042, Australia; adrian.linacre@flinders.edu.au

**Keywords:** animal forensics, wildlife forensics, cat STRs, dog STRs, cyt *b*, COI

## Abstract

Animal forensic genetics, where the focus is on non-human species, is broadly divided in two: domestic species and wildlife. When traces of a domestic species are relevant to a forensic investigation the question of species identification is less important, as the material comes from either a dog or a cat for instance, but more relevant may be the identification of the actual pet. Identification of a specific animal draws on similar methods to those used in human identification by using microsatellite markers. The use of cat short tandem repeats to link a cat hair to a particular cat paved the way for similar identification of dogs. Wildlife forensic science is becoming accepted as a recognised discipline. There is growing acceptance that the illegal trade in wildlife is having devasting effects on the numbers of iconic species. Loci on the mitochondrial genome are used to identify the most likely species present. Sequencing the whole locus may not be needed if specific bases can be targeted. There can be benefits of increased sensitivity using mitochondrial loci for species testing, but occasionally there is an issue if hybrids are present. The use of massively parallel DNA sequencing has a role in the identification of the ingredients of traditional medicines where studies found protected species to be present, and a potential role in future species assignments. Non-human animal forensic testing can play a key role in investigations provided that it is performed to the same standards as all other DNA profiling processes.

## 1. Introduction

The plight of endangered iconic species, such as tigers, rhino, and elephants, gains much media attention. Additionally, the affectionate bonds that humans have with domestic pets, particularly dogs and cats, is such that maltreatment of pets ranks highly in the human psyche. It may therefore seem contrary that the forensic investigation and prosecution of alleged cases involving animals (non-human) has always had a lower priority compared to human identification. As a further example regarding research in this area, there are often only a few talks in a miscellaneous section of the ISFG congress on DNA in wildlife crime and non-human DNA profiling compared to many days devoted to human identification. The reason for this low priority is many-fold, but a prime reason is that in the past it was perceived that investigations where the focus was on non-human material required lower standards than those accepted as the norm in human identification. It was the case that so much of non-human DNA typing has traditionally been performed in non-accredited laboratories, often at academic institutes, although this is now starting to change with a number of laboratories gaining full ISO17025 accreditation.

The fact that a review was requested as part of a Special Issue is further evidence of the continuing acceptance of including non-human DNA typing as a tool in forensic investigations. This review is in two main sections: section one looks at the parallels with identification of individual dogs and cats using processes akin to those used in human DNA profiling; the second section discusses the more traditional wildlife forensic testing where the key question is what species is present and hence the focus is on mitochondrial DNA sequence analyses.

## 2. Wildlife Forensic Laboratories

Genetic testing to enforce national legislation is rarely performed by the same operational laboratories that undertake human identification. This is primarily attributable to time and cost constraints, and the fact that most operational laboratories are accredited to undertake standard operating procedures relevant to human identification only. Most wildlife forensic investigations are undertaken by universities on an ad hoc basis and performed by university academics who may not routinely undertake work for the criminal justice system. Good laboratory practice has been encouraged, and guidelines have been suggested for Quality Control and Quality Assurance within an academic-based wildlife forensic laboratory [1]. There are notable exceptions. These include the US Fish & Wildlife Laboratory in Ashland, Oregon (https://www.fws.gov/lab/ (accessed on 12 March 2021)), the Scotland-based Wildlife DNA Forensics Laboratory (http://www.sasa.gov.uk/ (accessed on 12 March 2021)) and the Australian Museum in Sydney (http://www.Australianmuseum.com (accessed on 12 March 2021)); these laboratories have attained ISO17025 accreditation.

## 3. Short Tandem Repeat (STR) Analyses in Animal Forensic Science

Dogs and cats are two pet species that are commonly in close contact with humans. According to the North American Pet Insurance Association (NAPHIA), 65% of households in the USA, or 85 million homes, have a dog (https://www.iii.org/fact-statistic/facts-statistics-pet-statistics (accessed on 12 March 2021)). Equally over 9 million dogs and 8 million cats are owned in the UK (according to statista.com), this is about 25% of all homes. It is not surprising therefore that hairs from either dogs or cats are found associated with forensic evidence such as clothing, car seats, and household soft furnishing, an example is shown in Figure 1.

Such items onto which cat or dog hairs can be transferred, may themselves be associative evidence or be a substrate for secondary transfer to further items that may be of forensic relevance in an investigation. Therefore, it was with the first case of cat hairs to identify a person of interest and lead to a conviction.

### 3.1. Feline STR Typing

Snowball the cat became famous in arena of forensic DNA profiling for aiding in the resolution of an alleged homicide [2]. The work was initiated by the Royal Canadian Mounted Police (RCMP) after a 32-year-old woman from Prince Edward Island, Canada, was reported missing on 3 October 1994. This occurred in a remote wooded area. A man’s jacket was found close to the abandoned car of the missing person, along with other items. White hairs clinging to the lining of the jacket were identified by the RCMP’s laboratory to be from a cat. The only suspect in the case happened to own a white cat called Snowball.

Ten dinucleotide STR loci were used in a comparison of the DNA profiles from the hairs taken from the jacket to a hair samples that had been taken from Snowball; these ten loci were found to match. Now the issue was to determine the probability of this match and therefore a population genetic database was generated from cats in Prince Edward Island. In addition, a second database was generated from outbred cats from the Eastern United States [3]. The resulting match probabilities were 2.2 × 10^−8^ and 6.9 × 10^−7^ for the Prince Edward Island and the U.S. population databases, respectively [3].

The cat DNA evidence was accepted by the court, and in combination with additional human DNA evidence, was presented to the jury at the Supreme Court of Prince Edward Island. On 19 July 1996, the jury convicted the defendant of second-degree murder [2].

This first case was ground-breaking and showed the real potential of such evidence. The case required a number of steps: developing microsatellite markers, ensuring that the loci were variable, developing primer sets, ensuring the loci were not linked, testing to confirm that they could be co-amplified, developing an allelic database that was representative to allow for the use of robust statistics to support any matching DNA types. These steps are all part of the characterisation of hypervariable STR loci used in human identification, although the use of commercially available products circumvents this massive undertaking.

The first uses of STR loci used dinucleotide repeat markers, but these suffer from a number of stochastic effects, particularly increase in stutter peaks. Feline STR testing was developed further by a group in Germany who characterised 14 hypervariable STR markers, predominantly tetranucleotide repeats, and developed and allelic ladder [4]. These work adhered to the recommendations of the International Society for Forensic Genetics [1]. To allow initial use of these 14 STRs in casework, the authors surveyed 122 unrelated cats in the local area and created an allelic database. Another key point to this work is that, with careful primer design, all 14 STR loci produced amplicons smaller than 260 bases; an important point when working on DNA from hairs and therefore likely to be degraded. An important aspect of this work is that the authors isolated and sequenced each locus to report on the underlying repeat structure. This type of work not only shows a model for the future, but also allowed interlaboratory testing to be performed.

Since this first case, there has been extensive research into increasing the number of hypervariable loci for both cats and dogs.

### 3.2. Canine STR Typing

A group based in Portugal characterised nine STR loci, all of which were tetranucleotide repeats, which were combined into one multiplex PCR [5]. The primer design ensured that the largest allele was still smaller than 350 bases. The authors had also isolated and characterised each of the nine loci so that the repeat sequence could be reported accurately. The authors had access to 113 dog samples to test the multiplex for all the standard conditions as expected in human identification. In combination, and based on the 113 sample set, the authors reported a very high discriminating power (matching probability = 1.63/10^10^). This type of work was the foundation for further canine studies.

This assay was extended by two more loci and used in the identification of wolves (*Canis lupus*) [6]. Dogs (*Canis lupus familiaris*) are a subspecies of the grey wolf and therefore the authors surmised rightly that the same primers would work effectively on wolf as on dogs. Wolves in European come into conflict with humans and are frequently blamed for killing livestock or attacking humans. A case study looked at one such case where a male had received many injuries to the body and claimed to been attacked by a wolf [7]. DNA typing showed conclusively that the man was attacked by a guard dog and not a wolf, or wolf/dog hybrid, as claim; the male later confessed that this was the case [7].

The International Society for Animal Genetics (ISAG) had published a list of putative dog STRs: these comprised 21 dinucleotide loci and 3 tetra repeats. Due to the issues of increased stutters with dinucleotide repeats, and that tetra repeats were recommended by the ISFG [1], a group at UC Davies developed a canine STR test that used 15 unlinked STR loci combined into one multiplex, entitled DogFiler [8]. A full validation was performed in accordance with the recommendations of the Scientific Working Group for DNA Analysis Methods (SGDAM). The authors of this study sequenced alleles to characterise fully the repeat motifs and also created allelic ladders to allow genotyping. The publication of DogFiler set a precedent as to how to create an STR multiplex that meets the same standards as commercially available kits used in human identification.

Often only shed dog hairs are available for analyses. In such instances, amplification of all the loci to generate a full STR profile might not be possible. To overcome this problem, the same group developed three mini multiplexes by moving the primers closer to the amplicons, thus shortening the length of any allele amplified [9]. The result is that no locus is greater than 205 bases in length. Therefore, called Mini-DogFiler met the same standards as the full multiplex.

The CaDNAP group based at the Medical University of Innsbruck separately developed 13 hypervariable microsatellite loci into a multiplex and analysed 1184 dogs [10]. All the dogs were from either Germany, Austria or Switzerland (the so called DACH countries). CaDNAP is a working group of the ISFG and members of the CaDNAP group have extensive experience in human identification and very familiar with the requirements for full validation prior to publication and use in the criminal justice system. The equipment required is also that used in any operational forensic science laboratory performing human identification. The inclusion of such a large number of individual dogs from many breeds allowed the construction of a dog population sample database. The chance of two dogs sharing an allele identical by decent could be calculated allowing the same statistics to be used as performed in human identification.

### 3.3. Livestock

Cats and dogs are the predominant domestic pets. Livestock such as cattle, horses, sheep and camel, are also sometimes the subject of forensic investigation. The ISAG is prominent in standardising genetic testing for parentage (stud or pedigree analyses) that can also be used in criminal cases if required. Two examples are provided below where case reports have been published.

There are currently 16 cattle (*Bos taurus*) STR loci listed on a prominent website (https://strbase.nist.gov//cattleSTRs.htm (accessed on 15 March 2021)), all of which are dinucleotide repeats. There is a StockMarks^®^ available from ThermoFisher Scientific (Waltham, MA, USA) that includes 11 loci. A recent report extended the panel of STRs to 30 [11]. This is largely for parentage studies where many loci are needed for confidence in associating a bull with calves, but also highly useful in confirming meat in the food industry.

Sheep (*Ovis aries*) are the most commonly grazed species and breeds such as merino and mouflon are either highly prized, or protected. A case report used 16 STR loci, that there amplified in four separate multiplex reactions, to link a carcass of a mouflon to items sized from a vehicle [12]. This case occurred on the island of Sardinia, used Bayesian assignment modelling to support this genetic linkage.

### 3.4. Wildlife STR Typing

The trade in ivory is leading to the extinction of elephants from many countries in Africa. An elephant tusk is an over-grown tooth and therefore a potential source of trace DNA. Linking ivory seizures by DNA typing from this genetic material opens the possibility to identify the number of elephants slaughtered in any seizure of ivory samples; such knowledge will aid in forensic investigations, or at least in highlighting the extent of the illegal trade. A group at the University of Washington led by Professor Sam Wasser developed a series of STRs from the two species of African elephant [13,14]: the savannah species (*Loxodonta Africana)* and the forest dwelling species (*Loxodonta cyclotis)*. The concept behind the study was to identify the geographic location from which the elephants were poached. It would be expected that elephants with shared recent ancestors (i.e., distant family groups), will have more alleles in common at the STR loci tested than individual elephants with a more distant genetic heritage. Knowledge of the way in which matriarchal herds roam over either the savannahs or forests of Africa aid in understanding clusters of STR alleles within a broad geographic area. This pioneering work not only had to overcome the problems in creating a multiplex of STR loci, but also in being able to isolate sufficient quality DNA from tusks, particularly when the tusks may have been stored in poor conditions such that bacterial action might degrade the trace DNA present.

RhODIS^®^ is a comparable system for the linkage of rhino horns to that of elephants [15]. The RhODIS^®^ test is promoted by the University of Pretoria and is comprised of 12 STR loci plus a gender test. It was developed for the two species of rhino that still live in Africa: the white rhino (*Ceratotherium simum*) and the black rhino (*Diceros bicornis*). A potential for RhODIS^®^ is that samples from members of the last remaining white and black rhino in wildlife parks such as Kruger can be collected and then later matched to seized horns taken from poached rhino. The author of this review was a guest of a World Bank and TRAFFIC supported visit to Kruger national park in eastern South Africa to witness RhODIS^®^ in action. Figure 2 is an image of the author attending the poaching of a mother rhino, nearby was the corpse of the offspring; both killed for their horns.

A panel of STR markers have also been created for the one horned Indian rhino (*Rhinoceros**unicornis*) [16], as this species has also suffered extreme loss of numbers due to poaching for their horns. This work is part of the official RhODIS-India program. The authors of this paper used 14 STR markers—these were all dinucleotide repeats and far from ideal but had been characterised previously and therefore saved time and resources rather than trying to find and characterise new tetra-repeats [16].

As is likely to be the case for any species that has suffered a loss in numbers and is now breeding from a small set of individuals, the frequency of some alleles starts to either increase as homozygosity increases, and conversely some alleles will decrease or be lost. As reported in the paper on the Indian rhino, many of the STR loci only have 3 or 4 alleles. Despite this, a power of discrimination was reported as 1.4  ×  10^−9^ [16].

Elephant and rhino are iconic mammalian species whose body parts are highly traded internationally. Other mammalian species for which STR multiplexes have been developed include bears, wolves and badgers. As populations increase in the USA there is greater encroachment onto territories of the American black bear (*Ursus americanus*). Such human bear interactions can lead to injury, and on very rare occasions, death of the human, so tracking down the bear responsible is needed so that an innocent bear is not shot. Eleven tetra-nucleotide repeats were selected along with a sex test and made into one multiplex [17]. The test worked on trace amounts of DNA, down to 78 pg, so it is possible using this method to generate profiles from hairs and scat. A further example is that of STR typing for the European badger (*Meles meles*) [18]. Badgers are still hunted illegally and therefore the need to link samples from dead badgers or badger parts to items found in the cars, homes or persons of interest is needed.

Bird species are traded illegally for their use in the pet trade, i.e., parrots and cockatoos, or persecuted as perceived threats to livestock, i.e., eagles, harriers and hawks. Cockatoos can be brightly coloured and are also known to be birds that can be caged and then kept as pets. As so happens, a criminal case spurred this research as there was a need to analyse the parentage of seized cockatoos, resulting in the application of 20 STR loci to aid in resolving a dispute as to whether eggs in possession of a person of interest had been taken from a nest illegally [19]. Hen harriers (*Circus cyaneus*) were once a very common sight over northern European moorlands, but now they are extremely rare due to habitat loss and being shot or poisoned as a perceived threat to grouse and other game birds. In response to the need to link carcasses of dead birds to biological material (blood and feathers) on seized items (such as traps and snares) a multiplex of 8 STR markers was created (tetra-repeats with one tri-repeat locus) and validated for use [20]. A sex test based on the CHD 1 gene (male birds are Z,Z and females are W,Z) was included and an allelic ladder created.

Reptiles and amphibians have also been the subject of development of STR panels. The carpet python (*Morelia spilota*) is common in the pet trade and can be kept in captivity if bred from pairs that were also captive (i.e., not taken from the wild). However, the number of known captive snakes appears greater than the expected captive bred population and therefore it is likely that snakes are collected illegally from the wild. In response to a need to test this, three separate multiplexes that covered a total of 24 loci, plus allelic ladders for each locus, was created and called OzPythonPlex [21].

The illegal collection of animals from the wild to fuel the pet trade extends to tortoise. The slow-moving reptiles are commonly kept as pets as they are easy to look after. A result, along with habitat loss, is that the number of tortoises in the wild has fallen dramatically. An example is Herman’s tortoise (*Testudo hermanni*), that used to span much of the northern Mediterranean countries. To ensure that seized tortoise are correctly relocated to their correct geographical range, a panel of STR markers were applied for geographical provenance [22].

### 3.5. Finding, Characterising and Applying Short Tandem Repeat Loci

In all the above reports using STRs, there was an extensive amount of research required to develop a multiplex STR test. For those from the forensic community whose experience is solely in human identification, the arduous steps to develop such a multiplex assay are unlikely to be known. The author of this review is from a time when the first STR test was reported, using four STR loci [23], which then to six STRs plus amelogenin in what was called ‘second generation multiplex (or SGM) [24]; SGM was made in-house by the UK Forensic Science Service.

The use of STRs from dogs, elephant, rhino and hen harriers as examples, is based on previously reported putative loci. More recently, such as from pythons, the entire genome of the animal was sequenced using massively parallel DNA sequencing (MPS) technology [21]. From the mass of genetic sequence data, there are software programs to locate repetitive DNA, then provide a list of putative primers [25]. Having ordered primer sequences, these need to be tested on DNA from multiple members of the species to ensure reproducibility and variability at the locus. If still suitable, then homozygotes at each locus need to be sequenced to confirm the repeat motif. Ideally the primers should have similar binding temperatures (termed TMs) so that they can all the amplified in the same reaction. By characterising the loci, an allelic window can be created to determine the smallest and largest alleles likely to be encountered; this is needed to ensure that the alleles do not overlap and fit within a size limit using up to 5 dyes. Sampling many members of the population leads to the creation of allelic ladders, essential for accurate genotyping. This sample set might form the basis of a frequency data set (typically need up to 200 individuals from a population). Standard population genetic calculations need to be undertaken: examining loci for Hardy–Weinberg equilibrium, checking if there is any genetic linkage, and examining for power of discrimination. Validation steps, akin to those provided by SGDAM, include: specificity testing to ensure the primers only amplify from the species/genus for which they were designed; sensitivity tests to determine the limit of detection and effect of adding sub-optimal mass of DNA; stability to see the effects of environmental factors; reproducibility to see the effect of running multiple samples for instance down a column by capillary electrophoresis; interlaboratory testing to ensure the test works on varying equipment and by different operators; blind trial testing to ensure the results are not biased by the inadvertent selection of samples; and perhaps finally publishing the data, as was the case for so many of the papers cited in this review. It is clear reading the above, that this is not something achieved in a matter of weeks and also the reason why so few complex multiplex assays exist for species other than humans and domesticated species such as dogs and cats.

## 4. Species Assignment in Wildlife Forensic Science

A question raised in wildlife forensic science is ‘*which species is this?*’. This is because legislation at a national level is many countries prohibits the trade or ownership products originating from legally protected species. The illegal trade in protected species is central to much of wildlife forensic science as tools employed by forensic practitioners can be applied when called on to determine whether something is, or is not, legal to trade or own.

Interpol recognized wildlife crime as the second most prevalent crime worldwide after drug trafficking [26]. The value of the trade is often quoted as $20 billion (USD) per year [26,27] although the exact figure is unknown as very little illegal trade is ever detected. It is only from intercepting seizures that the true scale be estimated. Furthermore, the low penalties and potential large financial gain lead to extensive and lucrative trade in endangered species [28,29,30]. This figure of USD 20 billion is also only the estimated illegal revenue from international trade and does not include the illegal trade within countries.

The scope and range of wildlife crime is wide and encompasses a variety of criminal activities such as poaching and illegal hunting of mammals [31,32], birds [20,33] and reptiles [21,34,35], and the use of animal derivatives in traditional medicines [36,37].

Forensic investigations can only be undertaken if it is alleged that legislation has been breached. Further, the willingness to conduct DNA profiling of suspected illegally traded/owned samples requires: Approval from prosecuting authorities, availability of persons competent to conduct the work, and funds to compensate that laboratory performing any wildlife investigation.

### 4.1. Legislation Covering Wildlife Trade

The international trade in endangered species is monitored and regulated through recommendations made by Convention on International Trade in Endangered Species of Wild Fauna and Flora (CITES) [38]. These recommendations are enforced at a national level by legislation. Normally, this legislation stipulates the names of the species that are protected. It is for these reasons that much interest at a research level has focused on methods of species identification [38,39].

Many countries have legislation covering the collection from the wild, breeding, ownership or international trade in animal parts or whole organisms. An example repository of information detailing the federal law in the USA can be found at https://www.animallaw.info/article/federal-wildlife-law-20th-century (accessed on 15 March 2021). The US Endangered Species Act (ESA) of 1973 is a key piece of legislation for both domestic (USA), and international conservation. The act aims to provide a framework to conserve and protect endangered and threatened species and their habitats. Additionally, in the USA, the Lacey Act is central to conservation of wildlife and dates from 1900 when it became the first federal law protecting wildlife. The Act makes it unlawful to import, export, sell, acquire, or purchase fish, wildlife or plants that are taken, possessed, transported, or sold: 1) in violation of U.S. or Indian law, or 2) in interstate or foreign commerce involving any fish, wildlife, or plants taken possessed or sold in violation of State or foreign law. The Lacey Act covers all fish and wildlife and their parts or products, plants protected by CITES and those protected by State law; thus regulating international trade of protected species.

The EU, comprised currently of 27 countries, has the free movement of goods across internal borders as a core value. It is only therefore trade across the first border of a member state that is an issue. The EU Council Regulation (EC) No 338/97 deals with the protection of species of wild fauna and flora by regulating trade of any species covered by this regulation. It lays down the provisions for import, export and re-export as well as internal EU trade in specimens of species listed in its four Annexes.

In the UK, the Wildlife and Countryside act of 1981 gave protection to native species, controlled the release of non-native species, protected sites of special scientific interest, and made provision for rights-of-way. A devolved Scotland and separately administered Northern Ireland and Wales resulted in local legislation to protect the environment.

### 4.2. mtDNA in Species Testing

It may be that the seized material has been processed to create a sculpture, ornament, item of clothing, food, or supposed medicinal product. DNA can be obtained at trace levels from ivory [40] where ivory is made into statues, or rhino horn is proceesed into knife handles [32]. The most traded mammalian species currently is one of the members of the pangolin family. Pangolins are the only mammals to have scales on the outer surface and are one of eight species: three of which are critically endangered (*Manis culionensis*, *Manis pentadactyla* and *Manis javanica*), three are endangered (*Phataginus tricuspis*, *Manis crassicaudata* and *Smutsia gigantea*) and two (*Phataginus tetradactyla* and *Smutsia temminckii*) are recorded as being ‘vulnerable’ on the Red List of Threatened Species of the International Union for Conservation of Nature. The threat to pangolins is their misguided use in traditional medicines, specifically when the scales are used as bogus remedies. DNA can be obtained from pangolin scales, even when processed as a soup [41,42]. Performing mtDNA sequencing can link seizures of pangolins, but also indicates the geographical origin from where they were taken from the wild to inform authorities about hotspots of poaching.

Ivory, rhino horn and pangolin scales are examples of evidential material where DNA is at trace levels. This is typical of much of the seizures encountered in the illegal wildlife trade such that mtDNA is the only genetic material available [28]. Human identification has centered on a small section of mtDNA that does not encode; termed the hypervariable regions that constitute less than 1000 of the 16,569 bases that make up the human mitochondrial genome. Vertebrate mtDNA has two strands of different densities: the heavy (or H-strand) and the light (or L-strand). The mitochondrial genome in eukaryotes encodes a total of 37 genes, 22 of which encode tRNA molecules, two encode rRNA molecules, and the other 13 encode proteins involved primarily with the process of oxidative respiration [39]. While the number of these genes on the mitochondrial genome does not vary within all vertebrate mitochondrial genomes decoded to-date, the order of the genes may alter [39]. As an example, the order is different between avian and mammalian mitochondrial genomes.

Although the use of mitochondrial loci in species testing has many benefits, there are a few disadvantages as well. Part of the benefit of mitochondrial loci is that they are inherited maternally and as a haplotype with no recombination. Maternal inheritance can be a problem if hybrid species occur, particularly if the mother is from a nonprotected species and the father is from a species that is protected [43,44] (see Section 4.5).

### 4.3. Species Identification Using Loci on Mitochondrial Genome

A classic characteristic for a locus used in species identification is that it should exhibit very little variation for any member of the same species such that all members of that species have the same, or very nearly the same, sequence; this is referred to as intraspecies variation. The second characteristic is that this same locus should exhibit sufficient differences from any member of the next closest species, being interspecies DNA sequence variation.

The main locus used in taxonomic and phylogenetic studies until recently was cytochrome *b* (cyt *b*) [45,46]. This occurs between bases 14,747 and 15,887 in human mtDNA and encodes a protein 380 amino acids in length. The cyt *b* locus has been used extensively in taxonomic and forensic studies [32,34,35,47].

The cytochrome *c* oxidase I (COI) mitochondrial gene locus was adopted by the Barcode for Life Consortium (BOLD) (www.boldsystems.org (accessed on 15 March 2021)) [48,49,50]. COI is found between bases 5904 and 7445 in human mtDNA and at a similar position for all other mammals. COI was used initially in the identification of invertebrate species [51,52] and became the locus of choice in forensic entomology [53] before being adopted by BOLD.

Currently, there is no standardized locus in species testing, hence a divergence between cyt *b*, COI and other locus such as ND5. The Commission for the International Society for Forensic Genetics (ISFG) recommended that there should be some rationale behind the choice of the locus used in any analysis [1]. It is most likely that different loci will exhibit varying inter- and intraspecies similarity depending on whether examining insects, other invertebrates, fish, reptiles, birds, or mammals. Mammals are the only taxonomic group where a detailed study has been performed to determine whether cyt *b* or COI have fewer false positive and false negative identifications [54]; in this study, cyt *b* was found to outperform COI.

It may often be the case, such as for ivory and rhino horn, that the DNA is highly fragmented and therefore not possible to examine the entire locus of either COI or cyt *b*; rather, sections of the loci are used. The first 400 bases section of the cyt *b* gene is typically used [46]. For the COI the first 645 bp are used as part of BOLD [48]. The use of either section of the cyt *b* or COI loci is aided by the design of universal primers; these are primers that can be used in PCR to amplify a section of a gene from all the species from which the specific primers are designed. For instance, up-stream from the cyt *b* gene is a gene for a tRNA molecule, and this sequence is highly conserved such that the primers will bind to any mammalian species. This is a real benefit if the material is single source and not contaminated with DNA from another species; as an example, it is all too easy to amplify human DNA due to external contamination.

### 4.4. Mitochondrial Sequence Analysis

The standard workflow for species identification is to: (1) isolate DNA; (2) then quantify the amount of DNA isolated; (3) amplify the section of mtDNA chosen; (4) confirm that a PCR product has been generated; (5) purify the PCR amplicon; (6) DNA sequence the amplicon (in both directions); (7) compare the sequence data to a reference data base such as GenBank or ForCyt [55]. ForCyt was established to remove errors that do exist in GenBank [56]. When publishing the reference DNA sequence from a species to which comparisons of unknown material can be made, it is essential that this known sample comes from a voucher specimen; this is recommended by the ISFG Commission [1]. Software such as the Basic Local Alignment Search Tool (BLAST) compares the target sequence fragment to the data deposited on the database (GenBank) and produces a similarity score of the 100 most closely matched sequences. These are listed with the closest similarity at the top of the list. The ideal is a similarity score of 100% to a known species, with the next closest being less than a similarity of 95%. This situation rarely occurs with a similarity score of 98% between the sequence tested and the nearest sequence being more typical. There is no consensus on how many differences and over what number of bases is either due to intra- or interspecies variation. It was reported that one base variation per 400 bp sequence is acceptable intraspecies variation [39].

### 4.5. SNP Testing

Performing a complete DNA sequence analysis from a section of DNA provides the maximum amount of genetic information. It may be that only a few DNA bases that differ (are polymorphic), and that much of the DNA sequence is of little value as the species being considered share much of their DNA. In such instances, if the polymorphic DNA bases are known, these bases can be targeted specifically. A recent example of this is the identification of SNPs within the mitochondrial genome of tigers [57]. In this instance, small parts of the tiger mitochondrial genome are amplified in one reaction using primers specific for the genus *Panthera*. The assay is based on SNaPshot^®^, a process used in ancestry assignments of human populations [58,59,60]. As the sections amplified in the first instance are short, no more than 300 bases, this type of DNA testing works very effectively on highly degraded DNA, with a recent example being the species assignment from ivory [61]. In this paper the authors were able to differentiate ivory taken from Asian elephants (*Elephas maximus*) compared to either of the African elephants with a 99.92% reported accuracy based on an impressive 140 samples of ivory. The paper targeted SNPs in either the cyt *b* gene or ND5 gene. Additionally, because the SNP detection can be species-specific, the test works well even when there is a complex mixture of other species. An example was to identify up to 18 European mammalian species by using universal primers in the cyt *b* gene in combination with species specific primers [62]. The size of the resulting species-specific amplicon is itself specific for each species and with 3 amplicons per species there is redundancy to cope with one of the tests not working.

### 4.6. Hybrids

Sequence comparison using regions of the mitochondrial genome has a tremendous advantage over nuclear markers for sensitivity; this is due to the multiple copy number of mitochondrial genomes per cell. A potential disadvantage in species testing is the maternal inheritance of mitochondrial DNA. When the male is from a protected species, but the female is from species not protected, and viable young are produced, due to the maternal of inheritance of mtDNA, the offspring will not be recognised as a protected species. This issue was highlighted in a recent report [43], which is based on a talk at the ISFG congress in 2019 in Prague. There are in reality only a few examples where hybrids are the result of this type of mating as the majority of examples include species that are geographically isolated, or of little forensic relevance: examples of geographical separation are mating between an African lion (*Panthera leo*) and Indian tiger (*Panthera tigris tigris*) resulting in either a tigon or liger; and of limited forensic relevance between a horse (*Equus caballus)* and a donkey (*Equus asinus)* to produce a hinny or mule.

An example where hybrids are an issue was highlighted in a case report on the alleged poaching of protected South American camelids: llamas (*Lama glama*), vicuñas (*Vicugna vicugna*), alpacas (*Vicugna pacos*) and guanaco (*Lama guanicoe*) [44]. Wild (vicuña and guanaco) and domesticated (llama and alpaca) animals roam within Chile freely and can mate to produce viable young capable of themselves producing young. In Chile it is illegal to hunt and kill guanaco and llama but hybrids are not protected; this leads to the potential to sell meat from protected species as from a legally killed animal. The colour of the coats of the domestic and hybrids are different and specific to the variants. This led to the group in Chile, who were faced with this problem, to use in addition to cyt *b* sequence data, a diagnostic base in the melanocortin 1 receptor (*MC1R*). The combination of species assignment with a marker for coat colour resolved this case.

### 4.7. Traditional Medicines

Seizures of rhino horn, ivory and pangolins usually are of either identifiable using morphology or by DNA as will be from only one species. An example where multiple protected species are found in a complex mix, along with other plant and animal derivatives, is the trade in traditional medicines. Here, the ingredients might not be listed, rather the spurious cure for while they are taken can be prominent. An example is shown in Figure 3.

These plasters and pills list bear and tiger on the packaging, which is an offence in many countries, breaching CITES agreements, even if there is no tiger or bear present. It is likely that the amounts of any genetic material would be at extremely low amounts, and in a complex mix with potentially other species. This makes standard species testing problematic. Using universal primers and then amplifying using standard Sanger sequencing of part of the mtDNA will result in unreadable DNA sequence data. Undoubtedly the application of massively parallel DNA sequence technology overcomes the issue of any complex mixtures. The first example of this work on seized traditional medicines identified a number of protected species along with large numbers of other plant and animal species not listed on the packaging but with poor health outcomes [37]. A more recent study found protected species in traditional medicine products that could be bought in Australia [36], yet no prosecutions were undertaken even though there is clear evidence of breaching federal law.

## 5. Conclusions on Animal Forensic Profiling

This review clearly identifies the need, and the growing use of, non-human animal DNA for a range of forensic applications. There are a few laboratories dedicated to this type of analyses and can demonstrate that they work to the same exacting standards required in human identification. More such facilities are needed and, when it comes to wildlife forensic science, in countries closer to where seizures happen. The STR typing of non-human animals, with cats and dogs being prime examples, uses the same equipment as in human identification and therefore creating Standard Operating Procedures and adopting the STR typing of dog for example into use in mainstream accredited laboratories is therefore possible.

There remains no standard locus for species testing. A wealth of data exists on GenBank for sections of cyt *b*, COI and other mtDNA regions such as ND5 from a wide range of animal species. It may be that one locus has fewer false negatives and false positives than another for different taxonomic Orders, although data only exist in some mammalian species as a comparison. Intraspecies variation needs to be known for many more species as currently there are very little data available. One answer would be to use the power of MPS and analyse multiple loci in one reaction. The common use of MPS is on the near horizon as prices for consumables and the use of this equipment decreases. MPS is also seeing acceptance in other areas of non-human DNA typing such as in analyses of bacteria for post mortem interval [63,64] and now in tracking microbiomes [65,66] and was the subject of a recent review [67]. If the use of MPS is growing a profile in bacterial work, then there is every reason to expect the same acceptance and profile from eukaryotic and in particular vertebrate species.

The application of non-human DNA, now with examples of now bacterial DNA typing, can add an extra dimension to a forensic investigation. The use of DNA from species other than humans can add a further dimension to a forensic investigation. It is for those working in the non-human arena to ensure that their standards meet those expected and can provide a valuable service to the criminal justice system.

## Figures and Tables

**Figure 1 genes-12-00515-f001:**
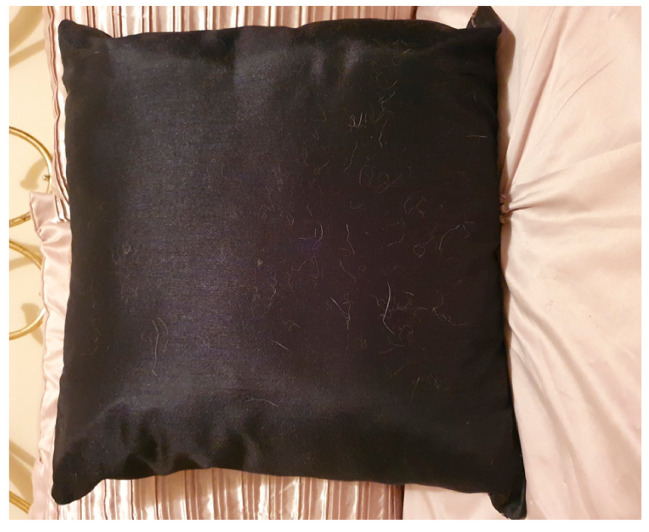
Showing how easy it is for cat hairs to transfer onto soft furnishing, in this case a cushion. These cat hairs can be valuable associative forensic evidence. Note also that these hairs can be transferred to another substrate by secondary transfer.

**Figure 2 genes-12-00515-f002:**
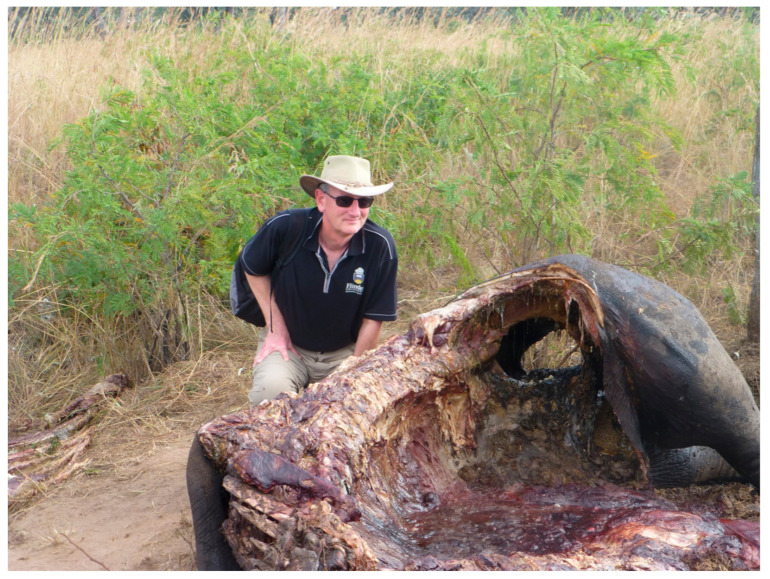
Showing the author at the site of a recently (within 4 days of taking the image) poached mother rhino. The rhino had been shot and then the horn removed with a chain saw. The image was taken in Kruger National Park, South Africa, and is one of around 34 poached rhino recorded in the park every month.

**Figure 3 genes-12-00515-f003:**
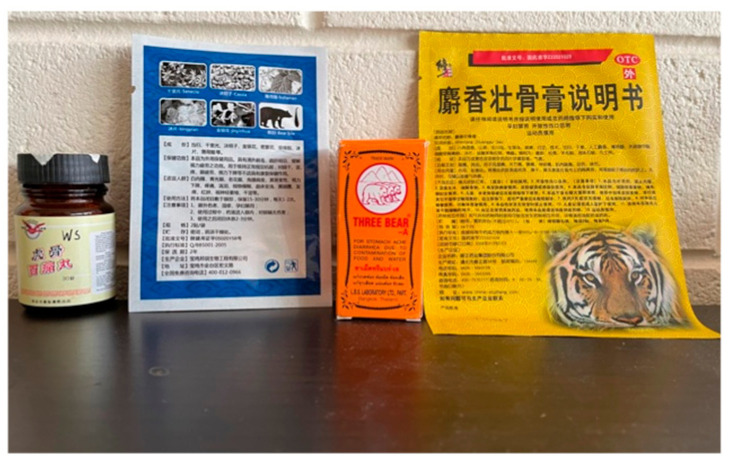
Showing examples of a traditional east Asian medicine that was seized by Border Force officials. The items were later show to contain traces of DNA from endangered species.

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
