# Peer review of "Animal Forensic Genetics"

_genes, 2021, doi:10.3390/genes12040515_

Round 1

Reviewer 1 Report

Animal Forensic Genetics

Summary:

The review summarizes significant aspects of the broad and diverse field of forensic animal genetics. The review is divided in two main parts, the first addresses the DNA-based individualisation of cats, dogs, and wildlife and the second section relates to species assignment mainly via mtDNA. Beside the molecular genetic aspects also legal regulations in different countries are mentioned.

Broad comments:

The text is written in a personal style (see e.g. Figure 2) and the author selected data from publications he considered as particularly significant. Therefore, the style of this review is not to mention as much as publications as possible. Rather, a selection of key publications is presented. One focus is the explanation of the chronological development (see e.g. chapter 3.1) and another is the description of technical and logistic challenges to implement applications for animal forensic genetics that meet the same high-quality standards as required in the field human DNA profiling. This makes the article informative for a wider readership. Generally, the concept of this review turned out well. Some improvement suggestions and minor revisions are listed in section "Specific Comments".

  • Originality/Novelty: It is a review, therefore no new (original) data are presented. The content is well prepared and helpful for interested readers
  • Significance: The presented (or summarized) data are interpreted appropriately. The author selected results of significance from the literature
  • Quality of Presentation: The review is written in an appropriate way and the content is presented in written form in a high standard (no Tables or Figures).
  • Scientific Soundness:  The soundness of the scientific content of the presented data from the literature is correct.
  • Interest to the Readers: This review can attract a wide readership within the forensic and animal genetic community as well as for people involved in animal/nature conservation.
  • Overall Merit: The author has addressed an important issue and has summarized different aspects of non-human forensic genetic .
  • English Level: appropriate and understandable

Specific Comments/Minor Revisions:

  • The division between domestic species and wildlife is reasonable. However, as domestic animals almost exclusively cats and dogs are mentioned. The author should include a (short) section where other species of domestic animals particularly livestock are mentioned.
  • Feline STR Typing: please add a more recent reference e.g. Schury et al. 2014, FSIGEN
  • Canine STR Typing: It would be welcome to include also the contribution of the Portuguese working group e.g. van Asch et al. 2009 Electrophoresis
  • In the species assignment section too little attention was given for cases where wildlife can cause accidents or killing of livestock. I am thinking particularly of the “wolf problematic” which induced recently a lot of public interest in Europe. It would be worth to consider this aspect.
  • Line 122: CaDNAP is an ISFG working group; the point of contact is at “Medical University of Innsbruck”
  • Line 359: “study” instead of “studied”

Author Response

The division between domestic species and wildlife is reasonable. However, as domestic animals almost exclusively cats and dogs are mentioned. The author should include a (short) section where other species of domestic animals particularly livestock are mentioned.

This is a good point and as such I have now included new section 3.1 Livestock: this can be found between lines 161 to 177.

Feline STR Typing: please add a more recent reference e.g. Schury et al. 2014, FSIGEN

This was an omission. It is possible that I have omitted a few others – this is not due to any favouritism rather I was trying to emphasis key papers that made changes in the processes of forensic animal genetics. This work definitely should have been included; please find between lines 103 to 116.

Canine STR Typing: It would be welcome to include also the contribution of the Portuguese working group e.g. van Asch et al. 2009 Electrophoresis

Again, a study that brought about a step change so should have been added with associated text; lines 118 to 125.

In the species assignment section too little attention was given for cases where wildlife can cause accidents or killing of livestock. I am thinking particularly of the “wolf problematic” which induced recently a lot of public interest in Europe. It would be worth to consider this aspect.

This point flows well after the dog STR work so has been added: lines 126 to 133.

Line 122: CaDNAP is an ISFG working group; the point of contact is at “Medical University of Innsbruck”

Thank you for noticing this and making sure that these details are correct. The changes are on lines 1560 and 153.

Line 359: “study” instead of “studied”

Thank you: change made

Reviewer 2 Report

I appreciate the topic of this review. It is very interesting and not many reviews have been written on this topic.

In the abstract, row 11, the meaning of abbreviation STR is missing. It is mentioned there for the first time so there should be added "short tandem repeat". 

Row 54- abbreviations QC and QA are not explained.  

Figure 1- Personally, I think that a better photo could be used. But if this one is kept, at least the computer mouse cursor on the right side of the picture should be deleted.

Author Response

In the abstract, row 11, the meaning of abbreviation STR is missing. It is mentioned there for the first time so there should be added "short tandem repeat". 
Agreed, and to avoid abbreviations in the abstract, this is in full.

Row 54- abbreviations QC and QA are not explained.  
I have written these in full to make sure there is no ambiguity: please see line 54.

Figure 1- Personally, I think that a better photo could be used. But if this one is kept, at least the computer mouse cursor on the right side of the picture should be deleted.

Not removing the cursor was an error! I appreciate that many know that cat/dog hairs are transferred easily and considered the merits of this figure. On reflection, I would prefer to keep it. I have removed the old version of this figure and inserted a new, with no cursor.
